# *RAC1*-Amplified and *RAC1*-A159V Hotspot-Mutated Head and Neck Cancer Sensitive to the Rac Inhibitor EHop-016 In Vivo: A Proof-of-Concept Study

**DOI:** 10.3390/cancers17030361

**Published:** 2025-01-23

**Authors:** Helen Hoi Yin Chan, Hoi-Lam Ngan, Yuen-Keng Ng, Chun-Ho Law, Peony Hiu Yan Poon, Ray Wai Wa Chan, Kwok-Fai Lau, Wenying Piao, Hui Li, Lan Wang, Jason Ying Kuen Chan, Yu-Xiong Su, Thomas Chun Kit Yeung, Eileen Wong, Angela Wing Tung Li, Krista Roberta Verhoeft, Yuchen Liu, Yukai He, Stephen Kwok-Wing Tsui, Gordon B. Mills, Vivian Wai Yan Lui

**Affiliations:** 1School of Biomedical Sciences, Faculty of Medicine, The Chinese University of Hong Kong, Hong Kong SAR, China; helenchanhy@link.cuhk.edu.hk (H.H.Y.C.); hlnganjason@link.cuhk.edu.hk (H.-L.N.); alexlawch@link.cuhk.edu.hk (C.-H.L.); peony.poon@uk-koeln.de (P.H.Y.P.); wenying.piao.chum@ssss.gouv.qc.ca (W.P.); lihui@link.cuhk.edu.hk (H.L.); lanwang@link.cuhk.edu.hk (L.W.); ckyeung@codexedge.com (T.C.K.Y.); eileen2004@gmail.com (E.W.); juliacliu@connect.hku.hk (Y.L.); kwtsui@cuhk.edu.hk (S.K.-W.T.); 2Georgia Cancer Center, Augusta University, Augusta, GA 30912, USA; yng@augusta.edu (Y.-K.N.); yhe@augusta.edu (Y.H.); 3Department of Medicine, Medical College of Georgia at Augusta University, Augusta, GA 30912, USA; 4School of Life Sciences, Faculty of Science, The Chinese University of Hong Kong, Hong Kong SAR, China; 1155016021@link.cuhk.edu.hk (R.W.W.C.); kflau@cuhk.edu.hk (K.-F.L.); 5Department of Otorhinolaryngology, Head & Neck Surgery, Faculty of Medicine, The Chinese University of Hong Kong, Hong Kong SAR, China; jasonchan@ent.cuhk.edu.hk; 6Department of Oral and Maxillofacial Surgery, Faculty of Dentistry, The University of Hong Kong, Hong Kong SAR, China; richsu@hku.hk; 7Department of Pharmacy and Pharmacology, Li Ka Shing Faculty of Medicine, University of Hong Kong, Hong Kong SAR, China; angelaliwt@gmail.com (A.W.T.L.); krista.verhoeft@kuleuven.be (K.R.V.); 8Department of Biomedical Sciences, Li Ka Shing Faculty of Medicine, University of Hong Kong, Hong Kong SAR, China; 9Department of Biochemistry and Molecular Biology, Medical College of Georgia at Augusta University, Augusta, GA 30912, USA; 10Division of Oncological Sciences, Knight Cancer Institute, Oregon Health and Sciences University, Portland, OR 97201, USA; millsg@ohsu.edu

**Keywords:** *RAC1*-A159V hotspot mutation, *RAC1* amplification, head and neck squamous cell carcinoma (HNSCC)

## Abstract

*RAC1* is an oncogene. As of today, the biological importance and potential druggability of *RAC1* genomic aberrations remain largely underexplored in head and neck cancer. Findings from this proof-of-principle study showed that *RAC1*-amplified head and neck cancer patient-derived xenografts (but not *RAC1*-diploidy tumors), and engineered *RAC1*-A159V HNSCC xenografts (but not engineered *RAC1*-P29S HNSCC xenografts) were both sensitive to Rac targeting in vivo with a preclinical Rac inhibitor, EHop-16. These findings provide evidence supportive of the precision drugging of *RAC1*-amp or *RAC1*-A159V-mutated head and neck cancer patients (~5% cases of HNSCC, based on the Cancer Genome Atlas (TCGA) data) with new Rac1 inhibitors in clinical settings.

## 1. Introduction

*Ras-related C3 botulinum toxin substrate 1* (*RAC1*) is an established oncogene. It is a member of the Ras superfamily of small guanosine triphosphatases (GTPases). It functions as a molecular switch, cycling between the GDP-bound state (inactive) and GTP-bound state (activated state), to dynamically regulate cell growth, actin/cytoskeleton re-organization, cell adhesion, motility, and vesicle transport, etc. Its downstream effectors and effector signaling include RhoA (Ras homolog family member A), PAKs (p21-activated kinases), PAR-6 (partitioning-defective 6), IQGAP1 (IQ motif containing GTPase activating Protein-1), the PI3K pathway, the WAVE regulatory complex (WRC)-ARP signaling, etc. [1,2,3,4,5].

Rac1 signaling can be activated by extracellular stimuli (e.g., growth factor receptor, integrins, etc.), mRNA overexpression (which can be caused, in part, by gene copy increases), and somatic mutation [1,6,7,8]. Findings in specific cancer types (e.g., colon, breast, melanoma) showed that Rac1 overexpression could promote cancer cell aggressiveness (e.g., adhesion, migration, invasion), reactive oxygen species (ROS) production, glucose metabolism, and stemness, as well as resistance to conventional chemo/radiation therapy [9,10,11,12,13,14,15]. As of today, most functional studies on *RAC1* mutations are focused on the melanoma-prevalent P29S hotspot mutation (found in ~5–8% of melanoma patients), even though other *RAC1* mutations have been reported in various cancers [16].

Head and neck squamous cell carcinoma (HNSCC) is an aggressive cancer with ~0.93 million new cases/year. Compared to other major cancers with such a high prevalence, HNSCC lacks effective systemic treatment options. Recent pan-cancer genomic findings from us and others showed that a newly noted *RAC1* mutation, namely, *RAC1*-A159V (NM_006908.4 of 192 amino acids; alternatively marked as A178V per recent TCGA annotation with reference to NM_018890.3 of 211 a.a., www.cbioportal.org), was a specific recurrent hotspot mutation predominantly found in HNSCC [17] and several other cancers including cervical, colorectal, and lung cancers and sarcoma. Furthermore, structural prediction postulates that A159V could retain in its GTP-bound (active) form for much longer than P29S, suggesting the need for the functional characterization of this likely hyperactive *RAC1* mutant in the context of HNSCC [18]. Yet, the biology of this *RAC1*-A159V hotspot mutation has not been carefully examined in the context of HNSCC.

Here, using data from TCGA-HNSCC (n = 510 patients, Firehose cohort, USA), we noted *RAC1* gene copy gain in ~37% (188/510) of cases, *RAC1* amplification in 2% (10/510) of cases, and *RAC1* mutation in 3% (15/510) of cases, totaling ~42% of HNSCC cases. Based on the global incidence rate of 0.93 million new cases/year, just *RAC1* somatic mutations and high-level gene amplification alone would account for ~5% (i.e., ~46,500) new cases/year worldwide. Thus, it is important to study the functional roles and therapeutic potential of *RAC1*-amplified and *RAC1*-mutated HNSCC. Using functional genomics, we demonstrated that all TCGA-HNSCC-relevant *RAC1* genomic aberrations, including *RAC1* gene copy increase, could significantly drive HNSCC tumoroid growth and/invasion, with A159V, P29S, and K116N mutants being the most potent drivers among all. Interestingly, transcriptomics analyses revealed that multiple *RAC1* mutations, as well as gene copy increase, could drive PI3K pathway activation in HNSCC cells. Particularly, proteomic data showed that the *RAC1*-A159V hotspot mutant was associated with the prominent intra-tumoral upregulation of phospho-RPS6(Ser235/236) in HNSCC patient tumors (TCGA). Multiple metastatic genes were upregulated by *RAC1* mutations. Lastly, our proof-of-principle findings in vivo indicated that *RAC1*-amplified and *RAC1*-A159V-mutated HNSCC could be potentially druggable with EHop-016, a preclinical Rac inhibitor. Lastly, *RAC1*-amplified melanoma and *RAC1*-A159V-mutated endometrial cancer may also be druggable with EHop-016. Therefore, in principle, *RAC1* genomic aberrations can be potentially harnessed for precision medicine development in HNSCC.

## 2. Materials and Methods

### 2.1. HNSCC Tumor Samples and Sequencing

Targeted sequencing was performed using the IonS5 platform and Ion Reporter (ThermoFisher Scientific, Waltham, MA, USA). Clinical ethics approvals were obtained from the Research Ethics Committee of the Hospital Authority (University of Hong Kong/Hong Kong East Cluster; Joint Chinese University of Hong Kong–New Territories East Cluster; Kowloon West Cluster), Hong Kong SAR.

### 2.2. Cancer Cell Lines, HNSCC Patient-Derived Cultures, and Plat A Cell Cultures

The HNSCC cell line PECAPJ41CLONED2 (RRID: CVCL_2680) was purchased from Sigma-Aldrich, USA. FaDu (RRID: CVCL_1218) was purchased ATCC, USA. The human melanoma cell line, MeWo (CVCL_0445), and endometrial adenocarcinoma cell lines, HEC6 (RRID: CVCL_2931) and HEC251 (RRID: CVCL_2927), were all purchased from the Japanese Collection of Research Biosources (JCRB) Cell Bank, Osaka, Japan. The Platinum-A (PLAT-A) retrovirus packaging cell line was purchased from Cell Biolabs, San Diego, CA, USA. PECAPJ41CLONED2 and FaDu cells were maintained in IMDM and DMEM, respectively, with 10% fetal bovine serum (FBS, HyClone, Logan, UT, USA) and 1% penicillin-streptomycin (Gibco, Waltham, MA, USA). MeWo, HEC6, and HEC251 were maintained in Eagle’s Minimum Essential Medium (EMEM) with 10% FBS and 1% Pen/Strep. All cells were obtained with original authentication. Early passages from commercial sources were used within 2 months of culture.

Human HNSCC patient-derived cultures (PDCs) were generated in-house by our laboratory from clinically and pathologically confirmed HNSCC cases. In brief, HNSCC tumors were subjected to light trypsinization, followed by subsequent growth in enriched M199/M105 medium with 10 µM of ROCK inhibitor (Y27632 dihydrochloride, Enzo Biochem Inc., Farmingdale, NY, USA) [19]. ROCK inhibitor was removed at ~passage 10 and beyond. QM43-PDC, QM47-PDC, T11-PDC, and T76-PDC were maintained in medium 199 (cat. #11150-059. Gibco, USA) and medium 105 (cat. #M6395, Sigma-Aldrich, St. Louis, MO, USA) in a 1:1 ratio with 10% FBS and 1% Pen/Strep supplemented with 1% of insulin, 0.0025% (*w*/*v*) of cholera toxin, and 0.0004% (*w*/*v*) of hydrocortisone (all from Sigma-Aldrich, USA). All 4 PDCs have been confirmed to be pan-cytokeratin-positive for their HNSCC origin. They have also been confirmed to be negative for human papillomavirus (HPV)-16/18 and negative for mycoplasma, both by polymerase-chain reaction (PCR). Furthermore, all PDCs were also authenticated by whole-exome sequencing for their human origin (WES; 40–50 M reads/sample by NovaSeq 6000, DNA Link, Seoul, Korea). DNA from each of the respective original patient tumors and blood was also subjected to WES (DNA Link, Korea), enabling exact variant calling of both somatic and germline mutational events, as well as gene copy variation of each PDC (using the human reference genome GRCg37(hg19), DNA Link, Korea). Regarding *RAC1* genomic aberrations, all 4 PDCs did not carry *RAC1* somatic mutations, and only T11-PDC and T76-PDC carried increases in *RAC1* gene copy. The PDCs were free of mycoplasma and free of cellular contamination from commercial cell lines, including HeLa (CLS Cat# 300194/p772_HeLa, RRID: CVCL_0030) and other HNSCC cell lines. All cell cultures were maintained in an incubator at 37 °C and 5% CO_2_.

### 2.3. Retroviral Infection

The retroviral Plat-A amphotropic expression system (Cell Biolabs, Inc., USA) was employed for ectopic expression of *RAC1*-wildtype and mutants into HNSCC cells. Plasmids pMXs-puro-*RAC1*-WT, -A159V, -P29S, -G15S, -K116N, and -N39S mutations were generated by site-directed mutagenesis with sequencing verification by GenScript, Piscataway, NJ, USA, and transfected into PLAT-A cells using Lipofectamine 3000 (Thermo Fisher Scientific, USA) to generate respective retroviruses. Retroviruses were filtered through a 0.45 μm mixed cellulose ester membrane filter to remove cell debris and subsequently used for transduction of PECAPJ41CLONED2 cells for 72 h. Retroviruses were then removed from PECAPJ41CLONED2 cells, which were then cultured for Western blot validation and functional experiments.

### 2.4. Knockdown of RAC1

SMARTpool: ON-TARGETplus human RAC1 siRNA was purchased from Dharmacon, USA. ON-TARGETplus non-targeting pool (Dharmacon, Dharmacon Lafayette, CO, USA) was used as negative control. Transfection of cells were performed using Lipofectamine 3000 (Thermo Fisher Scientific, USA).

### 2.5. GST-PAK1 Pulldown Assay and Western Blotting

GST-PAK1 pulldown assay was performed with CHO cell lysates expressing various *RAC1* mutants and GST-PAK1 PBD proteins pre-incubated with GSH resin (Genscript, USA). Protein lysates were loaded into 12% SDS-PAGE gel, followed by incubation of primary antibody overnight and secondary antibody for 1 h, and subsequent chemiluminescence development with autoradiography. Antibodies p-AKT, AKT, p-MEK, P70S6K, and PIK3CA were purchased from Cell Signaling Technologies, Danvers, MA, USA. Βeta-Actin antibody (cat. sc-69879, 1:3000) was purchased from Santa Cruz, Dallas, TX, USA. Secondary antibodies GOAT RABBIT-HRP (#170-6515) or GOAT MOUSE-HRP (#170-6516) were from Bio-Rad, Hercules, CA, USA.

### 2.6. Matrigel Invasion Assay

Engineered PECAPJ41CLONED2 expressing EGFP, RAC1-wiltype, *RAC1*-G15S, *RAC1*-P29S, *RAC1*-K116N, *RAC1*-A159V, and *RAC1*-N39S was plated at a density of 5 × 10^4^ cells in 250 μL serum-free medium into 24-well Matrigel^®^ Invasion Chambers with 8.0 μm PET Membrane (cat. #354480, Corning^®^, Corning, NY, USA). Complete culture medium with 10% FBS was added into wells of 24-well plates to serve as a chemoattractant. Next day, medium was discarded, and the chamber membrane was then stained with Three-Step Stain Kit (cat. #3300, Thermo Scientific/Epredia, Kalamazoo, MI, USA). Cells invaded through the PET membrane were, therefore, stained and counted for quantification at 10× magnification under light microscope with brightfield.

### 2.7. 3D Spheroid Formation Assay

Ten thousand engineered PECAPJ41CLONED2 cells expressing EGFP, *RAC1*-wiltype, *RAC1*-G15S, *RAC1*-P29S, *RAC1*-K116N, *RAC1*-A159V, or *RAC1*-N39S were plated into 24-well flat-bottom ultra-low-attachment plates for spheroid formation (cat. #3473, Corning, USA). Growth of spheroids was monitored by CELENA X High Content Imaging System (Logos Biosystems, Annandale, VA, USA). Bright-field images (10× magnification) were taken 96 h after cell plating. Diameters of spheroids were measured using Image J Version 1.54h (ImageJ, RRID:SCR_003070).

### 2.8. In Vivo Experiments

Animal ethics approvals were obtained via the University Animal Experimentation Ethics Committee of the Chinese University of Hong Kong. Currently, there is no commercially available *RAC1*-A159V- or P29S-mutated HNSCC cell line or PDX model for in vivo experimentation. Thus, for a proof-of-concept, we retrovirally engineered such mutations, as well as the *RAC1*-wildtype gene and *EGFP*-control gene, into PECAPJ41CLONED2, which is *RAC1*-diploid and non-mutated (CCLE-2019 dataset, www.cbioportal.org), for investigation of the potential effects of EHop-16 targeting on *RAC1*-altered HNSCC models in vivo. These engineered cells were inoculated heterotopically by subcutaneous inoculation into the flanks of nude mice for testing of EHop-16 sensitivity vs. vehicle control. We would like to acknowledge that xenograft models established in nude mice lacking mature T cells and bearing defects in B-cell development could potentially affect tumor development and reaction to treatment, which the current study was not able to assess or evaluate. Though the nude mice-xenograft model is a conventional model for initial evaluation of anticancer drug activity in vivo, the absence of the immune system in these models is a shortcoming of the study as the models cannot truly reflect patients’ effects to such a treatment. For HNSCC xenograft experiments, 2 × 10^6^ engineered PECAPJ41CLONED2 cells were subcutaneously inoculated into the flanks of 3–4-week-old female nude mice. Similarly, 2 × 10^6^ engineered MeWo cells and 2 × 10^6^ HEC6 cells were injected subcutaneously into the flanks of female nude mice for the establishment of heterotopic xenograft models of melanoma and endometrial cancer, respectively. When tumor size reached ~ 4 × 4 mm^2^, mice were then randomized into different treatment or vehicle control groups (2 tumors per mouse, 4 mice per group).

For PDX experiments, 1 × 1mm^2^ tumor pieces were subcutaneously implanted into the flanks of NOD scid gamma mice. After ~1 month of implantation, when tumor size reached ~4 × 4 mm^2^, mice were randomly grouped for treatments vs. vehicle control (2 tumors per mouse, 4 mice per group). In brief, 10 mg/kg of EHop-016 (MedChemExpress, Monmouth Junction, NJ, USA) was dissolved in 2% dimethyl sulfoxide solution (DMSO) + 30% PEG300 + 5% Tween 80 + ddH_2_O or vehicle control was administered 3 times/week by intraperitoneal (i.p.) injection. Tumor volumes were measured using a digital caliper and calculated by the equation length × width^2^/2. Mice were sacrificed and tumors were collected for immunohistochemistry staining at experimental endpoints.

### 2.9. Immunohistochemistry and TUNEL Assay

Tumors were fixed in 4% paraformaldehyde overnight, followed by processing with 70% ethanol, embedding, and microtome sectioning. Dewaxing was performed with xylene, followed by antigen retrieval with citrate buffer. Immunohistochemistry (IHC) was performed using VECTASTAIN Elite ABC Universal PLUS Kit Peroxidase (Horse Anti-Mouse/Rabbit IgG) (cat. PK-8200). Anti-cytokeratin mouse antibody (DAKO, Glostrup, Denmark, #M3515, 1:500) and purified mouse anti-Ki67 Clone B56 antibody (BD Pharmingen™, San Diego, CA, USA, #55060, 1:500) primary antibodies were used. TUNEL assay was performed using In Situ Cell Death Detection Kit, Fluorescein (Roche, Basel, Switzerland, #11684795910) according to manufacturer’s instructions. Fluorescent images were obtained with Nikon Ti-2e Microscope and the number of TUNEL-positive cells/field were counted.

### 2.10. RNA-Seq Analysis

RNA samples were obtained from engineered PECAPJ41CLONED2 expressing EGFP, *RAC1*-WT, *RAC1*-G15S, *RAC1*-P29S, *RAC1*-K116N, *RAC1*-A159V, and *RAC1*-N39S and sent for RNA-seq by DNA Link, Korea. Transcript per million (TPM) gene expression levels of each RNA samples were estimated for their aberrant cell signaling pathway activity using Pathway RespOnsive GENes for activity inference (PROGENy) method [20]. RNA-seq expression value of TPM + 1 was used as a mathematical transformation in order to include gene expression comparison for genes with zero values. Upon TPM + 1 transformation, fold change in expression of all genes of *RAC1* mutants was compared with that of EGFP control for metastatic gene analysis. GO enrichment analysis was performed with the clusterProfiler Version 4.9.2 (R package) and the bubble plot was generated with ggplot2 Version 3.4.2 (R package).

### 2.11. Statistical Analysis

ANOVA (for experiments with more than 3 groups) or non-parametric Mann–Whitney test, two-sided were performed using the GraphPad Prism Version 9, USA. For in vivo tumor growth comparisons between vehicle and treatment groups, unpaired *t*-test was performed. *p* < 0.05 is considered statistically significant.

### 2.12. Availability of Data and Material

Publicly available data generated by others were used in this study. These were TCGA PanCancer Atlas and TGCA HNSCC (Firehose Legacy) from the source www.cbioportal.org, Genomics of Drug Sensitivity Data (GDSC) from the source www.cancerrxgene.org, and copy number signatures and mutational signatures from the source Catalogue of Somatic Mutations in Cancer (COSMIC; https://cancer.sanger.ac.uk/cosmic (accessed on 1 July 2024)) or Depmap (https://depmap.org/portal/ (accessed on 1 July 2024)).

Data generated in this study are available within the article and within the Appendix A. Expression profile data generated by and analyzed in this study are included as Appendix A. Original data generated in this study are available upon request through the corresponding author.

### 2.13. Sex as a Biological Variable

For *RAC1* mutational analysis, we have included comparison between male and female patients of HNSCC and pan-cancers.

### 2.14. Inclusion and Exclusion Criteria

In this study regarding genomic data analysis, all TCGA adult subjects are included from TCGA Pan-Cancer Atlas and TCGA-HNSCC Firehose Legacy. Any genomic data from patients below the age of 18 were excluded from our analyses.

## 3. Results

### 3.1. Frequent RAC1 Gene Copy Increases Associated with Poor Patient Survival in Pan-Cancers and in HPV(−)HNSCC

The *RAC1* oncogene resides on chr7p.22.1, which has not been recognized as a common amplicon in human cancers. Yet, based on TCGA Pan-Cancer Atlas data, *RAC1* gene copy increases (amplification/gain) are found in 37% of pan-cancer cases (n = 10,967). In HNSCC, *RAC1* copy number increases are very common, accounting for 38.8% of cases [37% of cases with *RAC1* gain (188/510) and 2% of cases with *RAC1* amplification (10/510); TCGA-HNSCC Firehose], while heterozygous loss is only found in 4.7% of cases (24/510 cases). No *RAC1* homozygous deletion is found. Similar findings are observed in our in-house Asian HNSCC cohort with targeted sequencing [33.1% of cases (44/133) with *RAC1* copy increases and only 1 case with *RAC1* loss].

Kaplan–Meier survival analyses showed that patients with *RAC1*-amp/gain tumors (n = 4005) in the TCGA Pan-Cancer cohort had significantly poorer overall survival (OS) and progression-free survival (PFS) when compared with patients having *RAC1*-diploidy tumors (n = 6053; log-rank test, *p* < 0.05; Figure 1A). Specifically, for HNSCC, *RAC1* copy increases were also associated with poorer OS (*p* = 0.0005, log-rank test) and PFS in HNSCC (*p* = 0.0037, log-rank test) when compared to diploidy patients (Figure 1B, as indicated by red boxes underneath the *x*-axis, and Figure 1C). Note that *RAC1* copy increase was also significantly associated with poorer for OS in five other cancer types [glioblastoma multiforme (GBM), brain lower grade glioma (LGG), lung adenocarcinoma (LUAD), ovarian carcinoma (OV), and uterine corpus endometrial carcinoma (UCEC)], and poorer PFS in four cancer types [GBM, LGG, kidney renal papillary cell carcinoma (KIRP), and sarcoma (SARC)] in addition to HNSCC (Figure 1B, Appendix A). Interestingly, *RAC1*-amp/gain was also significantly associated with aneuploidy in 29 out of 32 cancer types, including HNSCC (Figure 1B, as indicated by orange boxes underneath the *x*-axis).

Subsequent clinicopathological analyses of HNSCC cases with *RAC1*-amp/gain (n = 206) showed that they were significantly associated with *TP53* mutation (*p* = 4.05 × 10^−7^ ****) and extracapsular spread (*p* = 0.004 **); both are characteristic features of aggressive HNSCC [21,22] (Appendix A). We further noted that *RAC1*-amp/gain HNSCC was associated with human papillomavirus (HPV) negativity with a very high statistical significance (*p* = 1.17 × 10^−8^ ****). Since HPV(+)HNSCC and HPV(−)HNSCCs are two distinct etiological subtypes of HNSCC, subsequent HPV stratification was performed for additional survival analyses. As shown in Appendix A and Appendix A, we found that *RAC1*-amp/gain status (vs. *RAC1*-diploid status) remained significantly associated with poorer OS (*p* = 0.0080 **, log-rank test) and DFS among HPV(−)HNSCC patients (*p* = 0.0403 *, log-rank test), whilst *RAC1*-amp/gain status was not associated with OS nor DFS among HPV(+)HNSCC patients (*p* = 0.3053, Appendix A). *RAC1* copy loss has no impact on HNSCC patient survival, irrespective of HPV-status (Appendix A).

To determine if somatic *RAC1* gene copy number would likely govern *RAC1* mRNA expression in pan-cancers including HNSCC tumors, we examined the correlation between *RAC1* gene copy number and its intra-tumoral mRNA expression level using bulk RNA-seq data. Consistently, in both TCGA pan-cancers and HNSCC datasets, *RAC1*-amp and *RAC1*-gain tumors displayed a 1.7-fold and a 1.2-fold increase in *RAC1* mRNA expression compared to diploidy tumors (Figure 1D, Appendix A), while *RAC1*-loss tumors have reduced intra-tumoral mRNA expression, implicating likely major roles of somatic copy number alterations in governing *RAC1* mRNA expression in human cancers.

### 3.2. Chromosome Instability in RAC1-Amp/Gain HNSCC

COSMIC has recently annotated 21 copy number (CN) signatures in order to help delineating etiologies underlying CN changes in cancer. Our CN signature analysis showed that *RAC1*-amp/gain HNSCC was associated with signatures of tetraploidy (increased signature CN2, *p* = 1.34 × 10^−5^ ****, reduced diploidy signature CN1, *p* = 5.40 × 10^−12^ ****), focal loss of heterozygosity (CN11, *p* = 0.0011 ***; CN12, *p* = 0.0027 **), homologous recombination deficiency (CN18, *p* = 0.006 **), tandem duplication (CN17, *p* = 0.0076 **), and decreased non-genome-doubling (CN9, *p* = 0.0082 **) (Appendix A). Overall, chromosomal instability may represent the underlying etiology of *RAC1*-amp/gain HNSCC [23], which is consistent with the aneuploidy nature of these tumors (Figure 1B).

### 3.3. RAC1 Mutations Distinctly Enriched in Melanoma and HNSCC Among Pan-Cancers

Analysis of TCGA Pan-Cancer Atlas dataset revealed a 9-fold and 4-fold enrichment of *RAC1* mutations in melanoma (6.53%, n = 29/444) and HNSCC (2.94%, n = 13/510), when compared to the average pan-cancer *RAC1* mutation rate of 0.728% (76/10433 cases). Comprehensive mapping of all reported *RAC1* mutations to date (from TCGA, AACR Project GENIE, COSMIC, and metastatic cancer data combined; www.cbioportal) reveals *RAC1*-A159V as a hotspot mutation mainly enriched in HNSCC (but also found in colorectal, cervical, and several other cancers, yet never in melanoma), while *RAC1*-P29S is a melanoma-prevalent mutation, which is also found in HNSCC and other cancers (Figure 1E, Appendix A). Similarly, for reasons unknown at the moment, we found that among pan-cancer cases, P29S mutations are largely male-predominant, while A159V mutations are, on the contrary, female-predominant (Figure 1E, Appendix A).

### 3.4. HNSCC-Relevant RAC1 Mutations Cluster Around GTP-Binding Domain in 3D

The GTP-binding domain (also called the G-box) of the human Rac1 protein comprises five conserved motifs, G1–G5. The amino acid (a.a.) A159 resides on the G5 motif (Figure 1E, mutation mapping), which makes direct contact with the guanine base to help distinguish guanine from other nucleotides to ensure substrate specificity. P29, on the other hand, does not reside on any of the five G-box motifs, but is situated nearby the G2 motif for γ-phosphate and Mg^2+^ ion interactions [24]. Using a 3D X-ray crystallography structure of the human Rac1 protein (PDB ID3TH5), we found that a.a. A159 and P29 were far from each other with a 9.5 Å distance apart, suggesting their likely differential effects (or magnitude of effects) on Rac1 activity (Figure 1E insert). Interestingly, except for N39, nearly all TCGA-HNSCC *RAC1* mutation sites (A159, K116, G15, and C18) cluster around the G-box in 3D, implying likely impacts on Rac1’s GTP-dependent functions.

### 3.5. A159V and G-Box Domain Mutations Associated with Poorer Outcomes in HNSCC Patients

Next, we evaluated the potential impact of *RAC1* mutations on HNSCC patient survival by Kaplan–Meier analyses. We found that A159V-mutated patients (n = 6) had poorer OS (12.48 vs. 56.44 months, *p* = 0.0296; Figure 1F), and DFS (20.99 vs. 67.74 months, *p* = 0.0436; Figure 1F) than those without. Notably, five out of these six A159V-mutated tumors are advanced HNSCC; all presented with perineural invasion (PNI) at diagnosis, indicating their aggressive nature. (Note: PNI records were only available for five of these patients; Appendix A). Also, considering patients with G-box-mutations (A159V, C18F/Y, and K116N/R), they were found to have poorer OS (27.04 vs. 56.44 months, *p* = 0.0440) and DFS (20.52 vs. 57.74 months, *p* = 0.0280) (Figure 1G) vs. those without such events. In fact, for reasons unclear, the majority of *RAC1*-mutated cases were HPV(−)HNSCC: Five out of the six A159V-mutated HNSCC cases (83.3%; 5/6) and eight of the 10 of the G-box-mutated cases (80%; 8/10) were HPV(−)HNSCC. We thus performed Kaplan–Meier analyses with HPV stratification. We found that either A159V- or G-box-mutation status alone was significantly associated with poorer DFS among HPV(−)HNSCC patients (Appendix A). Note that there was only one HPV(+)HNSCC patient with A159V mutation and two HPV(+) cases with any G-box mutations, which did not allow meaningful statistical analysis to be performed among HPV(+)HNSCC cases.

COSMIC mutational signature analysis showed that *RAC1*-A159V-mutated and G-box-mutated HNSCC tumors, when compared to non-mutated *RAC1*-wildtye (WT) tumors, carried a 2.3-fold enriched signature of SBS10b (*p* = 0.0128 for A159V; *p* = 0.043 for G-box-mutated HNSCC) associated with polymerase epsilon exonuclease domain mutations [20], and an 8.1-fold enriched signature of SBS20 (*p* = 0.0014 for A159V; *p* = 0.0284 for G-box-mutated HNSCC) associated with concurrent *POLD1* mutations, defective DNA mismatch repair, and microsatellite instability (MSI). Thus, defective DNA repair may represent a likely underlying etiology of *RAC1*-mutant HNSCC (Figure 1H).

In contrast, *RAC1*-P29S melanoma tumors do not carry any enriched mutational signature vs. WT-counterparts (*p* = n.s.).

### 3.6. A159V Represents the Most Potent Driver for HNSCC Growth and Invasion, More So than P29S

Next, we determined the potential effects of *RAC1*-A159V and the other HNSCC-relevant mutations on HNSCC cell growth and invasion using retroviral engineering (in isogenic HNSCC cell background). The HNSCC cell line, PECAPJ41CLONED2, is a *RAC1*-diploid and non-mutated cell line (CCLE-2019 dataset, www.cbioportal.org), which was deemed suitable for the evaluation of the functional effects of *RAC1* aberrations. [Note: currently, there is no commercially available HNSCC cell line bearing endogenous *RAC1* mutations for any biological or functional study in HNSCC]. We thus retrovirally engineered such mutations, including G15S, P29S, N39S, K116N, and A159V, as well as the *RAC1*-wildtype and *EGFP*-control genes, into the PECAPJ41CLONED2 cells and assessed the potential driver activity of these *RAC1* mutations with an isogenic cell background in PECAPJ41CLONED2. We found that in the 2D cell culture system, the overexpression of *RAC1*-WT (mimicking AMP) and G-box mutants (A159V, K116N) drove significant increases in cell proliferation vs. the EGFP control (*p* < 0.0001; Appendix A), while P29S, N39S, and G15S did not. Yet, in 3D tumoroid growth assay (with cells plated in flat-bottom ultra-low attachment condition), we found that *RAC1*-A159V possessed the strongest 3D tumoroid driver activity [average diameter = 198.6 pixel units (PU)], followed by K116N (178.7PU), G15S (153.4PU), P29S (152.1PU), N39S (144.4PU), and WT (134.4PU), when compared to the EGFP control (112.3PU) (Figure 2A). Similarly, *RAC1*-A159V was the most potent driver for 3D tumoroid growth in another HNSCC cell line, FaDu (Appendix A).

For the invasion assay, the ectopic expression of *RAC1*-A159V drove a 16-fold increase in cell invasiveness (*p* = 0.0002), which was seconded by K116N G-box-mutation (~9-fold increase, *p* = 0.0125), then by G15S and P29S mutations (~4-fold increase, *p* = 0.0065, and *p* = 0.0012, respectively) vs. EGFP-control cells (Figure 2B). For *RAC1*-WT overexpression (mimicking *RAC1* amp) and N39S, no change in invasion activity was observed. This was consistent with our 3D structural finding that N39 was located at a distance from other *RAC1* mutations (Figure 1E). Overall, A159V represents the most potent driver for 2D cell growth, anoikis-resistant growth, and invasion, more so than the well-known P29S, in HNSCC systems.

### 3.7. A159V Mutations Induces PI3K Signaling Activation in HNSCC

Using the available proteomic data of HNSCC patient tumors from The Cancer Proteomic Atlas (TCPA; www.cbioportal), we found that A159V-HNSCC patient tumors (n = 2, only two were available) had an average of a 3-fold increase in p-RPS6(Ser235/236) protein expression compared to non-mutated HNSCC tumors (n = 120; proteomic levels of 0.642 vs. −0.241; *p* < 0.0001 ***, Figure 2C). A similar analysis could not be performed for other *RAC1*-mutant tumors due to the lack of associated proteomic data. Yet, *RAC1*-induced PI3K pathway activation was also observed in engineered *RAC1*-mutant stable cells in PECAPJ41CLONED2 background (vs. EGFP- and *RAC1*-WT-expressing cells) as demonstrated by Western blotting. Figure 2D shows that different mutants could activate diverse PI3K nodes, including the upregulation of p-AKT (by A159V and G15S) and p-S6 (Ser235/236) (by all five mutants, except N39S). Note that among all mutants, A159V and G15S consistently upregulated both p-S6 (Ser235/236) (3.10-fold and 2.81-fold) and p-Akt (ser473) (2.65-fold and 2.83-fold), suggesting these mutants may engage AKT and p70S6K in promoting HNSCC tumorigenesis. Pak1 is an important effector of Rac1. We also found that the overexpression of both *RAC1*-WT (mimicking *RAC1*-amp) and *RAC1* mutants increased Pak1 levels by 1.5 to 2.6-fold vs. EGFP control (Appendix A).

Global transcriptome Pathway RespOnsive GENes for activity inference (PROGENy) analysis (20) on PECAPJ41CLONED2-*RAC1* mutant stable cells (vs. EGFP- and *RAC1*-WT-expressing cells) demonstrated perturbation of 14 important cancer signaling pathways (Figure 2E). Strikingly, activation of the PI3K pathway was observed across all *RAC1* aberrations, including *RAC1*-WT (mimicking *RAC1* amplification) by PROGENy analysis. Relative PI3K pathway activity score was the highest for P29S, followed by K116N, N39S, A159V, WT, and G15S (Figure 2E). Of the mutations assessed, P29S and A159V were identified as strong activators for both EGFR and MAPK pathways, which are key pathways driving HNSCC tumorigenesis [26,27,28] (Figure 2E). Moderate activation of EGFR and MAPK pathways was observed for the other two G-box mutants, namely, G15S and K116N. Unexpectedly, P29S, despite being frequent and highly transforming, was found to suppress hypoxia, p53, TNF-alpha, NF-kB, and VEGF signaling. Lastly, *RAC1*-WT overexpression increased VEGF (2.17-fold), hypoxia (4.41-fold), and TGF-beta signaling (10.59-fold).

### 3.8. Global Transcriptome Reveals Upregulation of Multiple Metastasis-Related Genes by RAC1 Mutants in HNSCC Cells

Global transcriptomic analysis with GO ontology showed distinct and marked functional enrichment for RNA splicing and spliceosome activity by *RAC1*-A159V as compared to *RAC1*-P29S-expressing PECAPJ41CLONED2 cells, as well as vs. *RAC1*-WT, suggestive of a likely increase in the requirement of global RNA processing by A159V (consistent with its potent driver activity) (Appendix A). Additional transcriptomic analyses using our in-house curated head and neck cancer-related metastasis gene list [29] revealed the upregulation of *adenylate cyclase 1* (*ADCY1*) by *RAC1*-WT and gain-of-function mutants (G15S, A159V, K116N, and P29S, but not N39S) (fold-change (FC) > 1.5 vs. EGFP control; Figure 2F). *ADCY1* overexpression has been shown to be involved in the migration, invasion, and metastasis of mucosal melanoma, which is commonly found in the mucosa of the head and neck region, followed by anorectal and genital mucosa [30]. P29S upregulated a large number of metastasis-related genes. In contrast, A159V had more restricted effects, which included the upregulation of *JMJD7-PLA2G4B* (*jumonji domain containing 7-phospholipase A2 group IVB*), a read-through transcript recently identified as a regulator of AKT signaling and HNSCC cell survival [31]. Consistent with this finding, *JMJD7-PLA2G4B* was upregulated in *RAC1*-mutated TCGA-HNSC patient tumors (*p* = 8.382 × 10^−4^; Figure 2G). Interestingly, *sphingosine kinase 1* (*SPHK1*), *thrombospondin 1* (*THBS1*), *cadherin 4* (*C4*), *noggin* (*NOG*), and *inhibin subunit beta A* (*INHBB*) were found to be upregulated by the three most potent invasion drivers we identified from our functional assay (i.e., A159V, P29S, and K116N) (Figure 2F). Specifically, *SPHK1* overexpression has been shown to increase HNSCC cell invasion, accompanied by increased expression of the matrix metalloproteinase-2 (MMP-2) and matrix metalloproteinase-9 (MMP-9) [26,27,32]. The extracellular matrix protein, THBS1, has been shown to be induced by TGFB1, to help promote HNSCC invasion through MMP and integrin signaling [28]. CDH4 is an intercellular adhesion molecule, which displays a tumor-specific upregulation in HNSCC (vs. adjacent normal tissue) [33]. Xie et al. showed that specific knockdown of *CDH4* by siRNA reduced HNSCC cell migration and invasion [25]. TGF-beta signaling contributes to HNSCC invasiveness and progression [34]. Both NOG and INHBB are regulators of TGF-beta signaling. Specifically, NOG is an inhibitor of bone-morphogenetic protein (BMP) signaling, including BMP-2, which has been shown to be involved in HNSCC invasion [35]. Based on our in-house curated aggressive gene analysis, NOG overexpression (RNA-seq, mRNA level > 1 z-score) was significantly associated with poorer PFS in HNSCC patients (*p* = 1.957 × 10^−3^ ***; TCGA Pan-Cancer Atlas, n = 515 cases; www.cbioportal.org), suggestive of its potential role in HNSCC progression. Lastly, INHBB overexpression has been found to be associated with advanced clinical staging and metastasis in oral squamous cell carcinoma (OSCC) patients [36].

### 3.9. Preferential EHop-016 Sensitivity in RAC1-Amp HNSCC Patient-Derived Cultures (PDCs) vs. RAC1-Diploid-PDCs

As of today, *RAC1* gene copy alteration remains undruggable in human cancer, including HNSCC. To test the hypothesis that *RAC1* gene copy increases could be potentially targetable in principle, we determined the sensitivity of several newly developed in-house HNSCC patient-derived tumor cultures (PDCs) that harbored different *RAC1* gene copy numbers, including *RAC1*-diploidy PDCs (QM47-PDC & QM43-PDC: CN = 2) and *RAC1*-gain (T11-PDC: CN = 3) and *RAC1*-amp (T76-PDC: CN = 4), towards a recently developed preclinical Rac inhibitor, EHop-016, in 3D culture conditions [37]. Using 3D CellTiterGlo assay, we found that T76-PDC was the most sensitive PDC to EHop-016 (*p* = 0.0006 ***; *RAC1*-amp), followed by T11-PDC (*p* = 0.0007 ***, *RAC1*-gain), while the *RAC1*-diploidy QM43-PDC and QM47-PDC were the least sensitive ones (Figure 3A, left). Consistently, results from our engineered PECAPJ41CLONED2 models also showed that stable cells overexpressing *RAC1*-WT (mimicking *RAC1*-amp status) were preferentially sensitive to EHop-016 vs. EGFP control in 3D cultures (*p* = 0.0323; Figure 3A, right). Thus, *RAC1*-amp HNSCC appears to be druggable, in principle, with preclinical Rac inhibitors. This is subsequently supported by data from an independent dataset, GDSC (Genomics of Drug Sensitivity in Cancer; https://www.cancerrxgene.org/), showing that the most Rac inhibitor-sensitive HNSCC cell line, HSC4, was a *RAC1*-amp HNSCC cell line with outlier sensitivity to EHT-1864, another Rac inhibitor in the GDSC2-HNSCC dataset. HSC-4 was ~10-fold more sensitive to EHT-1864 than other HNSCC cells (IC_50_ of HSC-4 = 3.99 μM vs. geometric mean value of all 37 GDSC-HNSCC lines = 35.5 μM; Appendix A). Interestingly, among GDSC cancer cell lines from other cancer types, EHT-1864 outliers also appeared to harbor *RAC1*-amp/gain, including stomach-ECC10 (Amp), ALL-BE13, Breast-OCUB-M, LUAD-NCI-H23, SCLC-DMS-53, and melanoma-A2058 (Gain), and three *RAC1*-mutated lines with copy increases (ECAD-KYSE-150-gain, Breast-DU-4475 with *RAC1*-P29S-gain, and melanoma-CHL-1 with *RAC1*-C18Y-amp) [38,39,40] (Appendix A). Thus, *RAC1*-amp/gain status may determine Rac inhibitor sensitivity in pan-cancers, including HNSCC.

Due to the lack of *RAC1*-mutated patient-derived models for HNSCC, we could only employ our engineered models to determine their potential drug sensitivity profiles in HNSCC. As shown in Figure 3A (right), among all of our engineered mutants in HNSCC background, *RAC1*-A159V-expressing cells displayed the highest EHop-016 sensitivity in 3D cultures, with an IC_50_ of 1537.2 nM, which was 2.95 times more sensitive to EGFP-control cells (IC_50_ = 4535.3 nM; n = 18 cumulative data from three independent experiments; *p* = 0.0102). Moderate increases in EHop-016 sensitivity were observed for K116N and P29S expressors, while G15S and N39S expressors were relatively insensitive vs. EGFP-control cells.

### 3.10. RAC1-Amp HNSCC-PDC Sensitive to EHop-016 with Increase in Cell Death

In addition to 3D viability assay (3D CellTiterGlo), we further confirmed the preferential growth inhibition of *RAC1*-amp T76-PDC vs. *RAC1*-diploidy QM47-PDC by exact viable cell counting (with quantitative high-content analysis of Hoechst-positive/ethidium homodimer-negative nuclei-counting assay) (Figure 3B, left). We found that this preferential EHop-016-induced growth inhibition of *RAC1*-amp T76-PDC was accompanied by a significant increase in cell death (vs. the diploidy QM47-PDC, *p* < 0.0001; Figure 3B, right; dead cells were stained red by ethidium homodimer staining).

Lastly, we showed that the specific knockdown of *RAC1* by siRNA in the *RAC1*-amp T76-PDC did result in a reduction in EHop-016 sensitivity (Figure 3C, cumulative data from four independent experiments; *p* = 0.0034 **, n = 20), while further forced overexpression of *RAC1* gene copy (by transient transfection of extra *RAC1*-WT gene copies) did further enhance its sensitivity towards EHop-016 by 21.05% (*p* = 0.0314 *, n = 20, Appendix A). Thus, the *RAC1* gene copy or level governs Rac inhibitor sensitivity in HNSCC patient-derived cultures.

### 3.11. RAC1-Amp HNSCC Patient-Derived Xenograft (PDX) Showed Marked Sensitivity to EHop-016 In Vivo, but Not RAC1-Diploidy PDX

Next, we determined the in vivo activity of EHop-016 on early passage (both passage 1) of the *RAC1*-amp PDX, T76-PDX, vs. *RAC1*-diploid QM42-PDX (a readily graftable and expandable in-house *RAC1*-diploidy PDX, as both QM47 and QM43 did not have graftable PDX available). Mice bearing these PDXs were treated with vehicle control (2% dimethyl sulfoxide solution (DMSO) + 30% PEG300 + 5% Tween 80 + ddH_2_O) or EHop-016 (10 mg/kg, 3 days/week) by intraperitoneal injection. As shown in Figure 3D, while the *RAC1*-diploid-QM42-PDXs (n = 8) were basically insensitive to EHop-016 throughout the entire treatment (right panel), the *RAC1*-amp-T76-PDXs (n = 7; left panel) were highly sensitive to EHop-016 with significant tumor inhibition observed as early as day 9, which lasted till the end of experiment, when compared to vehicle treatment. Further, such a marked growth inhibition by EHop-016 treatment was accompanied by apoptosis induction *RAC1*-amp-T76-PDXs as determined by TUNEL staining (Figure 3D bottom panel; *p* = 0.008; n = 4). No significant apoptosis was observed in *RAC1*-diploidy QM42-PDXs upon EHop-016 treatment. There was no apparent toxicity with EHop-016 treatment in vivo, as indicated by no change in the body weights of the EHop-016-treated mice vs. vehicle (Appendix A).

Our proof-of-principle finding with patient-derived models and engineered models demonstrated that *RAC1*-amplified HNSCC could be pharmacologically vulnerable to Rac inhibitor targeting, both in vitro and in vivo.

### 3.12. RAC1-A159V and RAC1-Amp Xenografts Sensitive to EHop-016 Sensitivity In Vivo

Based on all PDX resources or databases, there is no readily available *RAC1*-A159V-mutated HNSCC PDX for in vivo investigation. We thus employed retrovirally engineered PECAPJ41CLONED2 *RAC1*-A159V stable cells for a proof-of-principle drug sensitivity testing in vivo. Engineered PECAPJ41CLONED2-EGFP and *RAC1*-WT-overexpressing stable cells were also included as controls. Note that *RAC1*-WT overexpression mimics *RAC1*-amp status. These engineered cells were injected into nude mice for the establishment of xenografts for drug treatment. As shown in Figure 4A, PECAPJ41CLONED2-EGFP xenografts were insensitive to EHop-016. Importantly, xenografts overexpressing the *RAC1*-WT gene (mimicking *RAC1*-amp) displayed high EHop-016 sensitivity vs. vehicle treatment, consistent with our results with *RAC1*-amp HNSCC PDX above (Figure 3D; T76-PDX is *RAC1*-amp). Furthermore, consistent with our EHop-016 sensitivity findings with 3D spheroids in vitro (Figure 3A), *RAC1*-A159V xenografts were also sensitive to EHop-016, demonstrating significant tumor inhibition upon the third dose of EHop-16 treatment (*p* = 0.0281 *), and such an inhibitory effect lasted until the end of experiment on day 24 (n = 8; *p* = 0.0011 **). However, xenografts bearing the other driver mutation, namely, P29S, only showed minimal tumor growth inhibition by EHop-016 in vivo. Strikingly, the inhibition of the growth of A159V xenografts by EHop-016 was accompanied by a 9.04-fold increase in apoptosis (*p* = 0.0015 **; Figure 4B upper panel) and 64.26% reduction in Ki-67 expression (a cell proliferation marker) when compared to that of vehicle treatment (*p* = 0.0003 ***; Figure 4B lower panel). Similarly, upon EHop-16 treatment, *RAC1*-WT-overexpressed xenografts also showed significant apoptosis, which is accompanied by reduced proliferation as indicated by the reduction in Ki-67 staining, when compared to vehicle treatment. We noted that initial EHop-016 treatment did cause body weight reduction in mice, which was later resumed to normal (Appendix A).

As *RAC1* aberrations activate the PI3K pathway, we also noted that EHop-016 treatment was able to inhibit the PI3K pathway activation, as shown by reduced p-S6 ribosomal protein (Ser235/236) (84% reduction in *RAC1*-A159V mutant, 89% reduction in *RAC1*-WT expressors, respectively), and reduced PIK3CA expression (66% reduction in *RAC1*-A159V mutant, 93% reduction in *RAC1*-WT expressors, respectively). However, EHop-016 treatment only brought about moderate inhibition of the PI3K signaling proteins (24% and 23% inhibition of p-S6(Ser235/236) and total levels; Figure 4C) in *RAC1*-P29S-HNSCC xenografts, consistent with its minimal antitumor activity in vivo (Figure 4A).

In addition to HNSCC, *RAC1*-A159V hotspot mutation has also been identified in endometrioid carcinoma (TCGA-BG-A0MG-01, GENIE-MSK-P-0044987-T01-IM6, and GENIE-MSK-P-0067212-T01-IM7). The Cell Line Encyclopedia (CCLE) has published somatic mutational profiles of 1570 human cancer cell lines (CCLE-2019), which are publicly accessible via the cbioportal (www.cbioportal.org). Analysis of the CCLE-2019 cell line mutational profiles enabled us to identify an endometrioid cancer cell line, HEC6, that harbors an endogenous *RAC1*-A159V mutation. We thus obtained this cell line for testing of EHop-016 sensitivity both in vitro and in vivo. As shown in Figure 4D, HEC6 cells were significantly more sensitive to EHop-016 compared to HEC251 cells (a *RAC1*-WT diploid endometrioid cell line) (*p* = 0.0288, cumulative results from three independent experiments, n = 18). Lastly, HEC6 xenografts were also sensitive to EHop-016 vs. vehicle treatment (Appendix A). Thus, *RAC1*-A159V mutant cancers other than HNSCC may also be potentially druggable by *RAC1* targeting, in principle.

### 3.13. P29S, but Not RAC1 Amplification, Confers Sensitivity to Rac Targeting in Melanoma Models

In the TCGA melanoma dataset where primary tumors were sequenced, *RAC1*-amplification was found in 3.27% (12/367) of cases, whereas *RAC1* mutations were found in 5.4% (20/367) of cases, with the majority being the well-known melanoma-prevalent P29S hotspot mutation (TCGA-SKCM Firehose, n = 367). We examined the rates of *RAC1* amplification in several independent metastatic melanoma cohorts, as melanoma is known to be highly metastatic in nature. As shown in Figure 5A, *RAC1* amplification appeared to be more common in metastatic melanoma lesions in three independent cohorts, accounting for 21.875% of cases in MSK-NEJM 2014 (metastatic, n = 64) [41], 13.16% in metastatic melanoma (UCLA, 2016 (n = 38)) [42], and 6.36% in DFCI metastatic melanoma (n = 110; Science 2015) [43] than in primary melanoma.

Until now, preclinical Rac inhibitors all demonstrated limited activities in *RAC1*-driven melanoma models in vivo [16,44]. Also, no study has been conducted with EHop-016 treatment in melanoma models in vivo. Given our observation that *RAC1*-amplified HNSCC were pharmacologically vulnerable to EHop-016, we tested the hypothesis that *RAC1*-amp status in other cancer, such as melanoma, might also confer sensitivity to Rac1 targeting using EHop-016. As shown in Figure 5B,C, we found that MeWo-xenografts overexpressing the *RAC1*-WT gene (mimicking *RAC1* amplification) displayed sensitivity to EHop-016 targeting with significant tumor reduction, and apoptosis induction when compared to vehicle treatment (*p* = 0.0005 ***, n = 4), whereas MeWo-*RAC1*-P29S xenografts were basically insensitive to EHop-016 treatment (*p* = n.s., n = 4), which was consistent with our findings in HNSCC P29S-xenograft models. There was no significant change in body weights of mice upon EHop-16 treatment (Appendix A). Therefore, in addition to HNSCC, *RAC1* amplification in melanoma (representing 6–22% cases of metastatic melanoma) may also be targetable, in principle, by Rac inhibitors.

## 4. Discussion

As of today, *RAC1* aberrations remain chemically undruggable for any human cancers, including HNSCC. Here, we showed that *RAC1*-altered HNSCCs, particularly in HPV(−)HNSCC settings, are highly aggressive cancers that negatively impact patient clinical outcomes. Previously, no functional studies have been conducted on *RAC1* mutations or *RAC1* copy number changes in HNSCC settings, despite the recent realization of a new A159V hotspot mutation that is prevalent in HNSCC [17,18,45]. By functional genomics, we demonstrated that nearly all reported HNSCC-associated *RAC1* aberrations, including gene copy increases, and the HNSCC-prevalent *RAC1*-A159V mutation, as well as G15S, P29S, and K116N mutations (that sit at or around the G-box domain in 3D), were all potent drivers for HNSCC tumorigenesis. Specifically, our functional assays concluded that A159V was the strongest driver for HNSCC 2D and 3D tumoroid growth, as well as cellular invasion, more so than P29S and K116N mutations. These functional data help explain the recently noted unusual prevalence or enrichment of this new A159V hotspot mutation in HNSCCs, which are mostly advanced cases. Furthermore, *RAC1* mutations are associated with the increased expression of metastatic-associated genes in HNSCC, in both patient tumors and our engineered cell models. Lastly, findings in HNSCC-PDX, PDC, and engineered cell models showed that *RAC1*-amplification and A159V hotspot mutation, but not P29S, are, in principle, druggable with preclinical Rac targeting. EHop-016 treatment in *RAC1*-amp and A159V-expressing HNSCC tumors in vivo resulted in the inhibition of PI3K signaling. Given the fact that gene amplification has not constituted an avenue for precision medicine targeting for any gene in any cancer type yet, and the fact that *RAC1*-amp (2% cases) and A159V mutation (1.2% cases) in total (~3.2% cases) account for more than 32,960 HNSCC patients worldwide [1.03 million new cases/year by 2030 [46]], the precision targeting of this notable subset of patients would be clinically relevant and meaningful. Even specifically for HPV(−)HNSCC cases only, an estimate of HPV(−)HNSCC patients bearing *RAC1*-amp or A159V would be ~23,072, assuming ~70% cases of HNSCC are HPV(−)HNSCC. Therefore, based on the results of this proof-of-principle study, more clinically active Rac inhibitors should be developed and tested for their preclinical and clinical activities in HNSCC settings. Furthermore, it remains to be determined if *RAC1* aberrations are also involved in HPV(+)HNSCC tumorigenesis, and if so, if they are likewise druggable by chemical inhibitors or not. Though the current TCGA data only contain a limited number of HPV(+)HNSCC cases, the role of *RAC1* aberrations in HPV(+)HNSCC should warrant future investigation as the influence of the HPV(+)HNSCC endemics continues to rise globally [46].

Rac1 inhibitors are under active drug development [37,47,48,49,50,51,52]. Here, we demonstrated that, in principle, *RAC1*-amp and *RAC1*-A159V-mutated HNSCC were pharmacologically vulnerable to Rac inhibitor targeting with EHop-016 in vivo (in both PDXs and xenografts), with no apparent toxicity. Importantly, *RAC1*-diploidy HNSCC-PDXs were not sensitive to Rac inhibitor targeting in vivo. Extending our findings to other cancers, we also provided evidence that A159V mutation in endometrial cancer and *RAC1*-amp in melanoma (but not P29S melanoma hotspot mutation) may also be potentially druggable with Rac inhibitors.

Our findings that *RAC1*-amplification could be potentially druggable in HNSCC in vivo is novel. Yet, we would like to point out that with the current lack of publicly available/testable HNSCC models bearing endogenous *RAC1*-A159V mutation, our in vivo drug sensitivity finding using *RAC1*-A159V-engineered HNSCC models should only serve as a proof-of-concept finding to support future validation in HNSCC models with endogenous *RAC1*-A159V mutation, when they become available. Our additional findings on the in vivo Rac1 inhibitor sensitivity in *RAC1*-amp melanoma and *RAC1*-A159V-mutated endometrial cancer, and the noted outlier EHT-1864 sensitivity (inhibitor of Rac1, Rac2, and Rac3) from the GDSC dataset both suggest that the chemical targeting of Rac1 or Rac1, 2, or 3 may be worth more attention in other cancers, such as ECAD, stomach, ALL, breast, LUAD (based on GDSC data), and endometrial cancers, as well as melanoma (based on our findings).

New Rac GTPase inhibitors are currently under active preclinical/early clinical development, including EHT-1864 and NSC23766 [49,53], but have not yet been FDA-approved. EHop-016 is known to be >100-fold more potent than previous Rac1 inhibitors such as NSC23766 in metastatic breast cancer cells, and our data provide an opportunity for the further development of EHop-016 [37], or its analogs, for the precision drugging of aggressive *RAC1*-altered HNSCC. Lastly, based on our findings that G-box-clustering mutations of *RAC1* would impact both tumorigenic activity and the clinical outcomes of HNSCC, the development of a new class of inhibitors for some of these G-box mutants, in particular, A159V, may be clinically important as well. Lastly, the P29S mutant, which is situated at a distance from the GTP-binding site, may deserve separate attention for drug design, given that this hotspot mutation is not only found in melanoma, but also in HNSCC.

Previous studies suggested that Rac1 could be a potential therapeutic target for HNSCC, as evident by its tumor-specific overexpression, its elevated expression in relapsed vs. primary HNSCC [45], and its demonstrated contribution to HNSCC chemo-radioresistance [45], cell invasion (via both ΔNp63α/Rac1 and EGFR/Vav2/Rac1 axes) [22,54], adhesion and migration (via the RAPI-Rac1 axis) [21], etc. Here, our demonstration that *RAC1*-amplified and *RAC1*-A159V-mutated HNSCC could display preferential sensitivity to Rac inhibitor targeting in vivo further provides supportive evidence for an expansion of efforts in both preclinical and clinical development of Rac inhibitors for the treatment of HNSCC, as well as other cancers driven by *RAC1* aberrations.

## 5. Conclusions

In conclusion, our study revealed that *RAC1*-A159V mutation and several G-box mutations of *RAC1* were drivers capable of promoting HNSCC growth and invasiveness. Furthermore, we showed that *RAC1*-amplified and *RAC1*-A159V-mutant HNSCC could be potentially druggable with a preclinical Rac inhibitor, EHop-16. Our findings provide evidence for further clinical investigation of the potential sensitivity of *RAC1*-amplified and *RAC1*-A159V-mutant HNSCC to Rac inhibitors, when such inhibitors become clinically available for human trials.

## Figures and Tables

**Figure 1 cancers-17-00361-f001:**
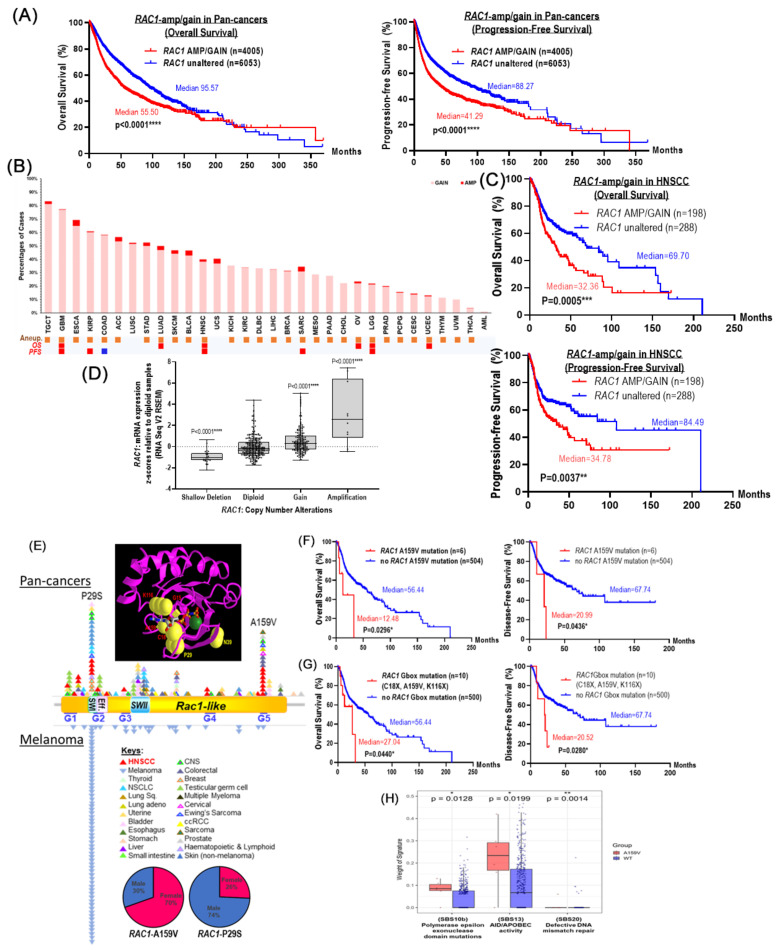
(**A**) Significantly poorer overall survival (OS; left panel) and progression-free survival (PFS; right panel) of pan-cancer patients with *RAC1*-amp/gain (n = 4005), versus those with unaltered *RAC1* copy number (i.e., diploid, n = 6053) in TCGA Pan-Cancer Atlas. Kaplan–Meier curves are shown with log-rank test *p*-values. Abbreviations—TGCT: testicular germ cell tumor; GBM: glioblastoma multiforme; ESCA: esophageal carcinoma; KIRP: kidney renal papillary cell carcinoma; COAD: colon adenocarcinoma; ACC: adrenocortical carcinoma; LUSC: lung squamous cell carcinoma; STAD: stomach adenocarcinoma; LUAD: lung adenocarcinoma; SKCM: skin cutaneous melanoma; BLCA: bladder urothelial carcinoma; HNSC: head and neck squamous cell carcinoma; UCS: uterine carcinosarcoma; KICH: kidney chromophobe; KIRC: kidney renal cell carcinoma; DLBC: diffuse large B-cell lymphoma; LIHC: liver hepatocellular carcinoma; BRCA: breast invasive carcinoma; SARC: sarcoma; MESA: mesothelioma; PAAD: pancreatic adenocarcinoma; CHOL: cholangiocarcinoma; OV: ovarian serous cystadenocarcinoma; LGG: lower-grade glioma; PRAD: prostate adenocarcinoma; PCPG: pheochromocytoma and paraganglioma; CESC: cervical squamous cell carcinoma and endocervical adenocarcinoma; UCEC: uterine corpus endometrial carcinoma; THYM: thymoma; UVM: uveal melanoma; THCA: thyroid carcinoma; AML: acute myeloid leukemia. (**B**) Percent cases with *RAC1*-amp/gain in 29 out of 31 TCGA pan-cancer types (n = 10967; except for LUSC and UCS) are shown. For each cancer type, % cases with *RAC1* gain (moderate increase, per TCGA definition) and amplification (high level increases, per TCGA definition) are indicated in pink and red, respectively. Orange boxes underneath indicated specific cancer types in which *RAC1*-amp/gain status is significantly associated with increases in aneuploidy score (abbreviated as Aneup.). Red boxes in overall survival (OS) indicate the 6 cancer types in which *RAC1*-amp/gain are associated with significant poorer OS (GBM, LUAD, HNSC, OV, LGG, and UCEC). Red boxes in progression-free survival (PFS) indicated the 5 cancer types in which *RAC1*-amp/gain are associated with significant poorer PFS (GBM, KIRP, HNSC, SARC, and LGG). (**C**) Significantly poorer OS and PFS of HNSCC patients with *RAC1*-amp/gain (n = 198) versus those with unaltered *RAC1* copy number (i.e., diploid, n = 288) in TCGA-HNSCC (Firehose n = 510). Kaplan–Meier curves are shown with log-rank test *p*-values. (**D**) Increase in *RAC1* mRNA expression in TCGA-HNSCC tumors with increasing *RAC1* copy number (TCGA-HNSC, n = 510). (**E**) **Top insert:** nearly all HNSCC-associated *RAC1* mutations located in close proximity of the GTP-binding site in 3D, where the chemical structure of the substrate, phosphoaminophosphonic acid-guanylate ester, is shown (source: X-ray crystallography structure of Rac1-wildtype protein; PDB: 3TH5). Amino acid residues K116, G15, A159, C18, P29, and N39, which are known to be mutated in TCGA-HNSCC tumors, are highlighted as yellow spheroids. Among these, K116, G15, A159, and C18 represent residues within the G-box (labeled in red), while P29 and N39 represent residues outside the G-box (labeled in yellow). **Middle:** Mapping of pan-cancer *RAC1* mutations on key functional domains of Rac1 protein. Each triangle represents a somatic mutation event identified in patient tumors from pan-cancer sources (curated from combined non-overlapping cases from cBioPortal, COSMIC, and GENIE databases). Mutation mapping of *RAC1* from pan-cancer (above) vs. melanoma (below) cases are shown separately. HNSCC-associated *RAC1* mutations are indicated as red triangles, and they appear to reside on key functional domains, especially at A159 position of G5-motif. Melanoma-associated *RAC1*-P29S mutations are indicated as blue triangles, which are mainly located on the switch II region. **Bottom:** gender distribution of two major *RAC1* somatic mutations in pan-cancers. *RAC1*-A159V-mutated pan-cancer cases are mostly found in females, while *RAC1*-P29S mutated cases are mostly found in male. (**F**) *RAC1*-A159V-mutated HNSCC patients and (**G**) G-box-mutated (C18X, A159V, and K116X) HNSCC patients have significantly reduced OS and DFS compared to *RAC1*-wildtype patients in TCGA-HNSCC Firehose cohort (n = 510 patients). Kaplan–Meier curves are shown with log-rank test *p*-values (* *p* < 0.05, ** *p* < 0.01, *** *p* < 0.001, and **** *p* <0.0001). (**H**) Mutational signatures associated with *RAC1*-A159V-mutated (red) vs. *RAC1*-WT TCGA-HNSCC tumors (purple). *RAC1*-A159V HNSCC tumors have significant increases in single base substitutions (SBS) mutational signatures of SBS10b (polymerase epsilon exonuclease domain mutations), SBS13 (AID/APOBEC activity), and SBS20 (defective DNA mismatch repair) vs. *RAC1*-WT HNSCC tumors (* *p* < 0.05, ** *p* < 0.01, *** *p* < 0.001, and **** *p* <0.0001).

**Figure 2 cancers-17-00361-f002:**
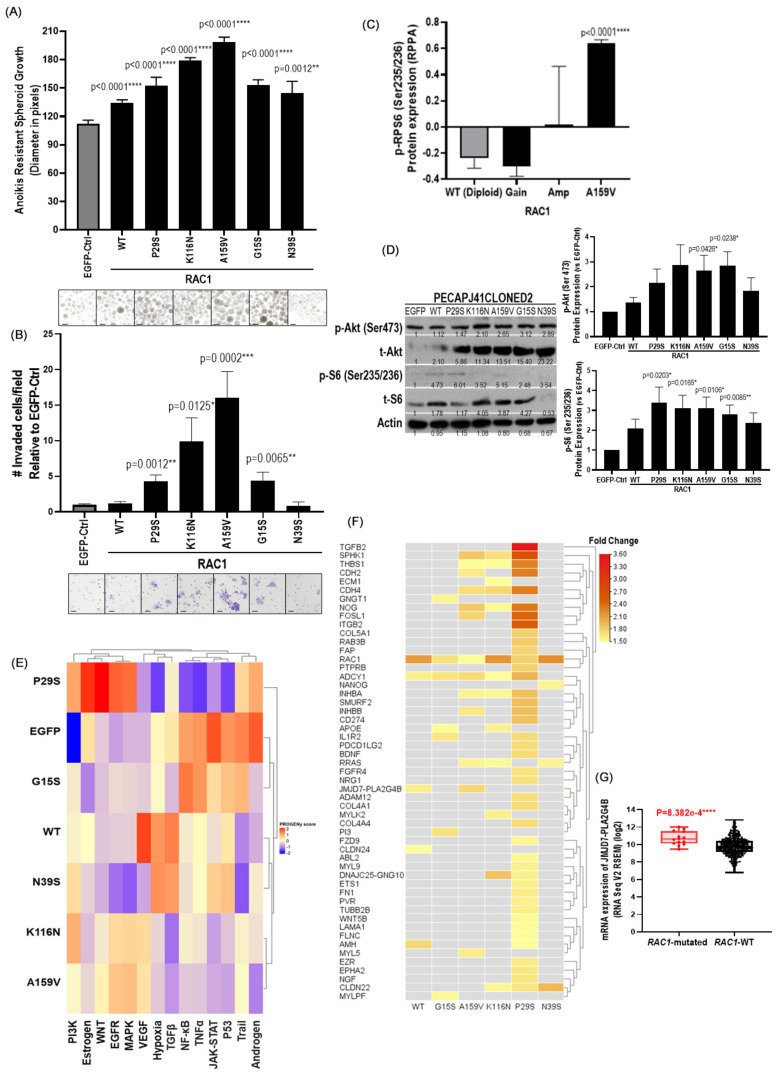
(**A**) Significant increases in 3D tumoroid growth of PECAPJ41CLONED2 stable cells expressing all *RAC1* aberrations (*RAC1*-WT, *RAC1*-A159V, *RAC1*-K116N, *RAC1*-G15S, *RAC1*-P29S, and *RAC1*-N39S) vs. EGFP control (n = 3 wells; 10,000 cells per well; * *p* < 0.05, ** *p* < 0.01, *** *p* < 0.001, and **** *p* < 0.0001). Cumulative results from >3 independent repeats were shown. Ten bright-field images were taken by CELENA X High Content Imaging System at 96 h after cell plating. A scale bar of 100 µm is shown. (**B**) Significant increases in invasiveness of PECAPJ41CLONED2 stable cells expressing *RAC1*-A159V, *RAC1*-K116N, *RAC1*-G15S, and *RAC1*-P29S vs. EGFP-expressing cells (* *p* < 0.05, ** *p* < 0.01, *** *p* < 0.001, and **** *p* < 0.0001). Graph showing cumulative results from >3 independent experiments in Matrigel^®^ Invasion Chambers with 8.0 μm pores using 5 × 10^4^ engineered cells. Bright-field images (10× magnification) at 24 h of invasion are shown. A scale bar of 100 µm is shown. (**C**) *RAC1*-A159V-mutated HNSCC patient tumors have significant increase of phospho-RPS6 (Ser235/236) protein expression compared with *RAC1*-WT (diploid) tumors in TCGA-HNSCC reverse-phase protein array dataset (n = 209). Normalized quantified values from the reverse phase protein array data are plotted (* *p* <0.05, ** *p* < 0.01, *** *p* < 0.001, and **** *p* < 0.0001). (**D**) Validation of PI3K pathway activation by HNSCC-associated *RAC1* aberrations by Western blotting in PECAPJ41CLONED2 stable cells expressing EGFP, *RAC1*-WT, *RAC1*-A159V, *RAC1*-K116N, *RAC1*-G15S, *RAC1*-P29S, and *RAC1*-N39S. Graphs showing cumulative results from >4 independent experiments with quantified fold changes of p-AKT(S473) and p-S6 (S235/236) levels vs. EGFP control (* *p* < 0.05, ** *p* < 0.01, *** *p* < 0.001, and **** *p* < 0.0001). The uncropped blots are shown in Appendix A. (**E**) Pathway RespOnsive GENes for activity inference (PROGENy) analysis showing universal PI3K activation in PECAPJ41CLONED2 stable cells expressing all HNSCC-associated *RAC1* aberrations vs. EGFP-control cells. Heatmap visualization of z-coefficients matrix for all features in the 14 cancer signaling pathways using ComplexHeatmap (R package, version 2.15.3). Activity score values ranging from −2 to 2 are displayed. (**F**) Heatmap showing significant up-regulation of metastasis-related genes by various *RAC1* mutations [fold change > 1.5 (TPM + 1) vs. PECAPJ41CLONED2-EGFP-expressing cells] based on our in-house curated HNSCC-metastasis-related gene list with 1283 genes [25]. (**G**) Significant increase in *JMJD7-PLA2G4B* mRNA expression was observed in *RAC1*-mutated HNSCC tumors vs. *RAC1*-WT tumors in TCGA-HNSCC cohort (n = 510; * *p* < 0.05, ** *p* < 0.01, *** *p* < 0.001, and **** *p* < 0.0001).

**Figure 3 cancers-17-00361-f003:**
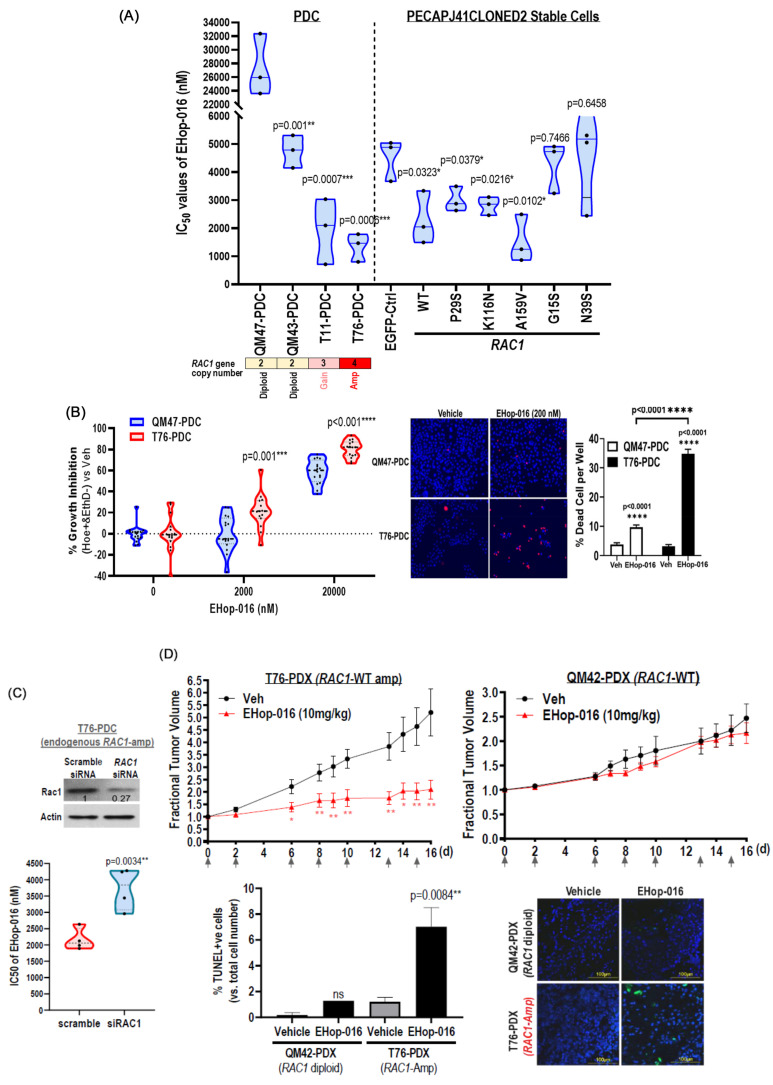
(**A**) HNSCC patient-derived cultures (PDCs) with *RAC1* gene copy gain and amplification [T11-PDC (*RAC1*-gain (3 copies): indicated by pink box) and T76-PDC (*RAC1*-amp (4 copies): indicated by red box)] have significantly lower IC_50_ for EHop-016 compared to *RAC1* diploidy PDCs [QM47-PDC (diploid), QM43-PDC (diploid); indicated by yellow boxes] (**Left**). Consistent results in engineered PECAPJ41CLONED2 stable cells showing *RAC1*-WT- and *RAC1*-A159V-expressing cells (as well as *RAC1*-K116N- and *RAC1*-P29S-expressing cells) have significantly lower EHop-016 IC_50_ than EGFP-control cells (**Right**). Sensitive cell lines are indicated in pink or red while insensitive cell lines are indicated in blue in the violin plot. (**B**) Preferential EHop-016-induced growth inhibition of T76-PDC (*RAC1*-amplified) vs. QM47-PDC (diploid) using the high content growth inhibition assay [percentage viable cells = (Hoechst-positive and Ethidium Homodimer-negative cells) divided by Hoechst-positive cells)]. PDCs were treated with EHop-016 for 24 h at 2000 and 20,000 nM concentrations, and then stained with Hoechst stain (all cell nuclei) and Ethidium Homodimer (dead cells). Images were taken by CELENA X High Content Imaging System (10× Bright-field magnification) and analyzed using CELENA X Cell Analyzer (Logos Biosystems, Annandale, VA, USA). Similar results were obtained from 4 independent experiments (n = 16 wells in total). (**C**) Specific downregulation of *RAC1* by *RAC1* siRNA in *RAC1*-amp T76-PDC resulted in reduced EHop-016 sensitivity, indicated by significant increase in IC_50_ (assayed by CellTiterGlo assay after 72 h of EHop-016 treatment) (**left**). *p*-values calculated by *t*-test (n = 4 independent experiments). The insert shows RAC1 protein levels of siRNA-transfected T76-PDC cells. Further increase in *RAC1* gene copy by *RAC1*-WT transfection in *RAC1*-amplified T76-PDC further promotes EHop-016 sensitivity, indicated by significant reduction in IC_50_ value for EHop-016 (assayed by CellTiterGlo assay after 72 h of EHop-016 treatment) (**right**). *p*-value calculated by *t*-test (n ≥ 3 independent experiments). The insert shows *RAC1* protein levels of *RAC1*-transfected T76-PDC cells. The uncropped blots are shown in Appendix A. (**D**) HNSCC PDX with endogenous *RAC1* amplification, T76-PDX, was highly sensitive to EHop-016 in vivo, while *RAC1*-diploid PDX, QM42-PDX, was not responsive to EHop-016 treatment in nod scid gamma mice. Arrows indicated the days of drug treatment. Fractional tumor growth of PDXs treated with vehicle or EHop-016 (10 mg/kg, 3 times a week, i.p.) are shown as mean with SEM (n = 8 per group). The bottom panel shows TUNEL staining for apoptosis induced by EHop-016 in T76-PDX, with quantification (n = 6 per group). Hoechst staining of nuclei (blue) was shown. Experiments were repeated at least 3 times independently with similar results. Asterisks indicated levels of statistical significance in all sub-panels above: * *p* < 0.05, ** *p* < 0.01, *** *p* <0.001, and **** *p* < 0.0001.

**Figure 4 cancers-17-00361-f004:**
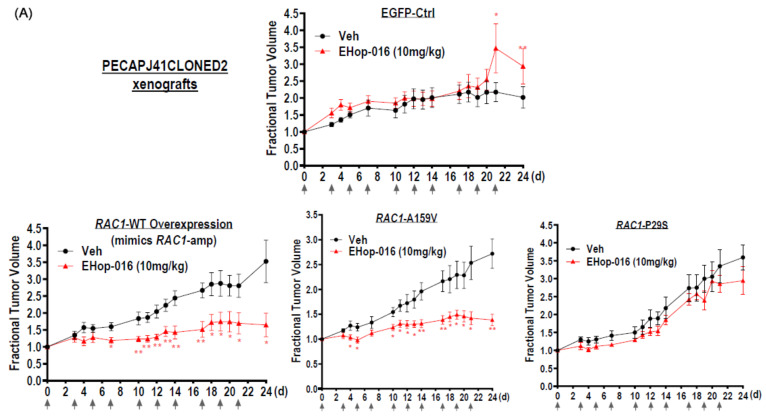
(**A**) HNSCC PECAPJ41CLONED2 xenograft models overexpressing *RAC1*-WT (mimicking *RAC1* amplification) and *RAC1*-A159V were highly sensitive to EHop-016 in vivo, while EGFP-control and *RAC1*-P29S xenografts were not responsive to EHop-016 treatment. Fractional tumor growth of mice treated with vehicle or EHop-016 (10 mg/kg, 3 times a week, i.p.) are shown as mean with SEM (n = 8 per group). (**B**) EHop-016 resulted in marked reduction of cell proliferation in *RAC1*-WT (mimicking *RAC1* amplification) and *RAC1*-A159V tumors (but not in *RAC1*-P29S tumors), as indicated by reduced Ki-67 staining, and with significant induction of apoptosis (as indicated by TUNEL-positive cell percentage). A scale bar of 100 µm is shown. (**C**) Specific downregulation of PI3K signaling by EHop-016 (vs. vehicle control) in HNSCC xenografts overexpressing *RAC1*-A159V and *RAC1*-WT (mimicking *RAC1* amplification), as indicated by reduction in PIK3CA, p-S6 (Ser235/236) and S6 protein expression. Experiments were repeated 3 times independently with similar results. The uncropped blots are shown in Appendix A. (**D**) Endometrial cancer cells, HEC6 (endogenous *RAC1*-A159V mutation; www.cbioportal.org), but not HEC251 (*RAC1*-WT diploid), displayed preferential sensitivity to EHop-016, as indicated by significantly lower IC_50_ of EHop-016 in HEC6 than in HEC251 (assayed by MTT at 72 h of treatment). Cumulative violin plots of IC_50_ of 3 independent experiments. *p*-value calculated by *t*-test. Asterisks indicated levels of statistical significance in all sub-panels above: * *p* < 0.05, ** *p* < 0.01, and *** *p* < 0.001.

**Figure 5 cancers-17-00361-f005:**
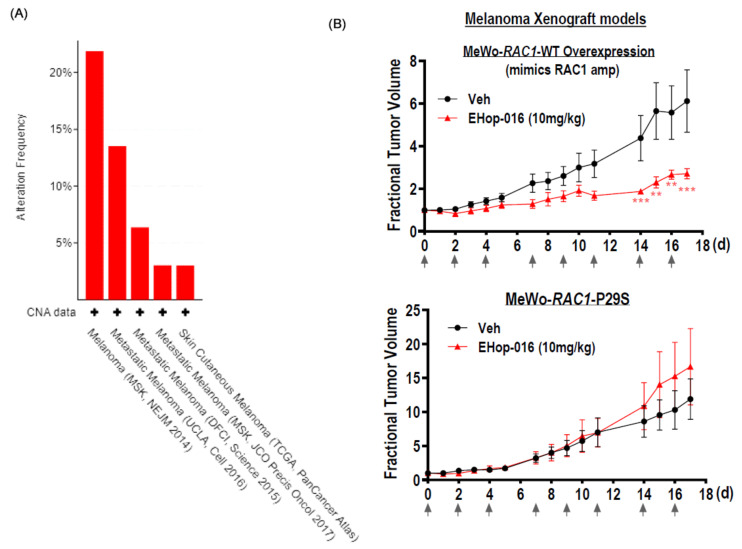
(**A**) Copy number alteration frequency of *RAC1* in metastatic melanoma in MSK, UCLA, and TCGA PanCancer Atlas cohorts. Source: www.cbioportal.org (data accessed on 10 May 2024). (**B**) Melanoma xenografts, MeWo-overexpressing *RAC1*-WT (mimicking *RAC1*-amplification), were sensitive to EHop-016 treatment (vs. vehicle control), while MeWo-*RAC1*-WT xenografts were not (upper panel). Fractional tumor growth of xenografts during treatment was shown (mean with SEM; n = 4 per group). In vivo EHop-016 sensitivity in MeWo-*RAC1*-WT xenografts was accompanied by significant induction of apoptosis (TUNEL), while no significant change in apoptosis was observed in MeWo-overexpressing *RAC1*-P29 (lower panel). (**C**) EHop-016 resulted in marked induction of apoptosis in *RAC1*-WT-amp MeWo xenografts when compared to vehicle treatment (*p* = 0.0005; by TUNEL assay, red dots indicated apoptotic cells), but not in *RAC1*-P29S MeWo xenografts (*p* = n.s.). Asterisks indicated levels of statistical significance in all sub-panels above: ** *p* < 0.01, and *** *p* <0.001.

## Data Availability

The data presented in this study including TCGA Pan-Cancer Atlas and TCGA-HNSCC (Firehose Legacy) from the source www.cbioportal.org; Genomics of Drug Sensitivity Data (GDSC) from the source www.cancerrxgene.org; Copy number signatures and mutational signatures were from the source Catalogue Of Somatic Mutations in Cancer (COSMIC; https://cancer.sanger.ac.uk/cosmic (accessed on 1 July 2024)), or Depmap (https://depmap.org/portal/ (accessed on 1 July 2024)). Data generated in this study are available within the article and its Appendix A. Expression profile data generated by and analyzed in this study are included as Appendix A. Original data generated in this study are available upon request through the corresponding author.

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
