# Peer review of "RAC1-Amplified and RAC1-A159V Hotspot-Mutated Head and Neck Cancer Sensitive to the Rac Inhibitor EHop-016 In Vivo: A Proof-of-Concept Study"

_cancers, 2025, doi:10.3390/cancers17030361_

Round 1

Reviewer 1 Report

Comments and Suggestions for Authors

The present paper describes RAC1 gene alterations, particularly A159V hotspot mutations, as a drug-able target in vitro and in vivo. It is very rich in original data, well written and abundantly illustrated.

The following comments might help to improve the manuscript.

About Materials and Methods.

One. The cell lines used need a more precise description as to origin and identification.  L111: Were cell lines that are generated in the authors laboratory also tested for cell type and origin? HEC6, HEC251, MeWo and FaDu cells are not described. “We were able to find” (L 610) is too vague a description of the HEC6 that harbors endogenous RAC1-A159V mutation . 

Two. The relevance of the xenograft model (L176) should be discussed. Nude mice lack components of the immune system that are crucial for tumor development and reaction to treatment. Implantation is heterotopic.  Are all cell lines (HNSCC, HEC, MeWo) handled in the same manner? 

About Figures.

Figure 1 B, 1 E and 1 H  are confusing and not well explained in the legend. Fig. 1 D: what do the colors indicate?

Figure 2A and B need scale bars.

Figure 3A: colors for gene copy number and violin plot are not well explained in the legend.

Figure 4D: replace Endometrial models by Endometrial cancer cell lines.

About details.

L73: add systemic, as  there exist excellent treatment options (radiotherapy, surgery) for HNSCC.

L 251: explain all abbreviations when first used. Also, in figure legends (L261).

L446 and reference 32: Briefly provide evidence for the relationship of these genes with metastasis. 

L558 & 660 “ for the first time” is not polite and possibly not true.

Reviewer 2 Report

Comments and Suggestions for Authors

The current manuscript “RAC1-amplified and RAC1-A159V Hotspot Mutated Head and Neck Cancer Sensitive to Rac Inhibitor In Vivo” intents to demonstrate potential of RAC-1 aberrations as a potential drug target against HNSCC. The authors used EHop-016 as a standard RAC inhibitor to reverse the HNSCC cell lines primarily derived from cancer specimens of HNSCC patients. The authors started with literature study of RAC-1 as a potential therapeutic target pan cancer. However, we are not convinced with RAC-1 as target against HNSCC based on the literature study (as cited below). RAC-1 is primarily a target in melanoma, colorectal and breast cancers. On the experimental design, the authors first used HNSCC patients’ specimens to derive cell lines, however, it would have been clearer if they could have identified naturally occurring relevant mutations in the cell lines, and worked on modulation of those cell lines. Instead, the authors used literature studies to screen specific types of mutations, and then induce those mutations in the PDCs, which again is not confirmed if the specifically designed mutations were established or not. In summary, we have apprehensions that the authors could not naturally justify the research gap of targeting RAC-1 mutation to reverse HNSCC proliferation. However, we request the authors to respond to the following queries before drawing any conclusions.

1.       How significant is RAC1 aberration in HNSCC in terms of clinical database? RAC1 is significant in some cancers but not others. Majority of the HNSCC does not have RAC1 aberration as major causal factor. A database http://dna00.bio.kyutech.ac.jp/PrognoScan-cgi/PrognoScan.cgi can be taken as reference. Kindly give a justification for its potential as a prominent target for HNSCC.

2.       Please provide the relevance of line 63-70 in the context of HNSCC. We think that the introduction may be streamlined a little.

3.       Kindly provide the evidence for 40,000 cases per year (line 80). Line 82, the authors have mentioned that RAC1 is amplified/gained in 42% of HNSCC, however, the cbioportal data suggest only 4.21% of 523 cases i.e., 22 cases (13 cases of mutation; 9 cases of amplification). Kindly provide a clarification.

4.       Since the authors claim that EHop-016 is a standard RAC inhibitor, what is the relevance of EHop-016 inhibiting RAC1 overexpression in melanoma and endometrial (?), in HNSCC manuscript?

5.       Since EHop-016 is the therapeutic molecule to study reversal of RAC1 overexpression in HNSCC, why have the authors restrained from using the inhibitor’s name in the title of the manuscript?

6.       What is the significance of knockdown RAC1, Matrigel invasion assay?

7.       Kindly recheck the superscripts of various numbers throughout the manuscript.

8.       Line 179, why have the authors mentioned “respectively”?

9.       Although, the authors have mentioned the inclusion/exclusion criteria and gender-based RAC1 mutational information, nowhere in the methodology we could identify these parameters while designing the experiments. We request the authors to kindly enlighten us on this observation.

10.   Figure 1, we fail to find relevance of data of expression of RAC1 in pan cancer in HNSCC manuscript. Figure 1B x-axis refers to different types of cancer gene mutations? We request the authors to provide clarity on our observation.

11.   Since all the mutations and copy number were generated in the cell lines derived from HNSCC patients, RAC1 mutations are not a normal phenomenon which could be used as a potential therapeutic target against HNSCC. Kindly respond to this critical observation.

12.   Kindly explain the relevance of figure 5 in the context of HNSCC.

Comments on the Quality of English Language

English requires significant modifications.

Round 2

Reviewer 2 Report

Comments and Suggestions for Authors

The manuscript has been significantly modified in contrast to the previous version. The authors have satisfactorily responded to all the major points categorically. However, with regards to query no. 11, the authors have humbly accepted the limitations in the experimental design and has satisfactorily responded to the same as “proof-of-concept”. In our humble view, it may be more appropriate to highlight the “proof of concept” in the title, lest the readers may be misled into believing it otherwise. There is a vast difference in establishing evidence for therapeutic mechanism against a cancer model and simply highlighting a “proof of concept”.

Author Response

Comment: The manuscript has been significantly modified in contrast to the previous version. The authors have satisfactorily responded to all the major points categorically. However, with regards to query no. 11, the authors have humbly accepted the limitations in the experimental design and has satisfactorily responded to the same as “proof-of-concept”. In our humble view, it may be more appropriate to highlight the “proof of concept” in the title, lest the readers may be misled into believing it otherwise. There is a vast difference in establishing evidence for therapeutic mechanism against a cancer model and simply highlighting a “proof of concept”.

Reply: Thank you very much for the reviewer's suggestion. We have amended the title accordingly in this revised manuscript entitled "RAC1-amplified and RAC1-A159V Hotspot Mutated Head and Neck Cancer Sensitive to the Rac Inhibitor EHop-016 In Vivo: A Proof-of-Concept Study"

We have also made some minor edits including L90, L494, L539, L603 and L772 (all highlighted in red). We also noted Fig.3B was omitted during editing in the previous revised version, which is now added back to the right place in this revised version. Thank you very much!